behaviour/cognition/psychology

magic effects, psychological constraints, cognitive evolution, corvid cognition, violation of expectations

**Author for correspondence:**
Alexandra K. Schnell
e-mail: alex.k.schnell@gmail.com

# Jays are sensitive to cognitive illusions

Alexandra K. Schnell[1], Maria Loconsole[1,2], Elias Garcia-Pelegrin[1], Clive Wilkins[1] and Nicola S. Clayton[1]

[1]Department of Psychology, University of Cambridge, Cambridge, UK
[2]Department of General Psychology, University of Padua, Padua, Italy

 AKS, 0000-0001-9223-0724; ML, 0000-0002-0024-2670; NSC, 0000-0003-1835-423X

Jays hide food caches, steal them from conspecifics and use tactics to minimize cache theft. Jays are sensitive to the content of their own caches, retrieving items depending on their preferences and the perishability of the cached item. Whether jays impose the same content sensitivity when they steal caches is less clear. We adapted the 'cups-and-balls' magic routine, creating a cognitive illusion to test whether jays are sensitive to the (i) content of hidden items and (ii) type of displacement. Subjects were presented with two conditions in which hidden food was consistent with their expectations; and two conditions in which food was manipulated to violate their expectations by switching their second preferred food for their preferred food (up-value) or vice versa (de-value). Subjects readily accepted food when it was consistent with their expectations but were more likely to re-inspect the baited cup and alternative cup when their expectations were violated. In the de-value condition, jays exhibited longer latencies to consume the food and often rejected it. Dominant subjects were more likely to reject the food, suggesting that social factors influence their responses to cognitive illusions. Using cognitive illusions offers innovative avenues for investigating the psychological constraints in diverse animal minds.

## 1. Introduction

Jays hide food in different locations for future consumption and not only remember the location of their own caches but also the location of caches that belong to other individuals. Jays rely on observational spatial memory to retrieve their own caches and to steal the caches of others [1–4]. To protect their caches from theft, jays employ a range of different cognitive tactics that limit the observer from obtaining visual or acoustic cues that might reveal the location of the store [5–7]. For example, jays preferentially cache in shaded sites or behind barriers to reduce

the quality and transfer of visual information to potential thieves [8–10]. Jays also prefer to cache in quiet substrates to conceal auditory information, particularly when a competitor cannot see the cacher but is within earshot [11,12]. In instances where a competitor's sensory access cannot be reduced, cachers misdirect potential pilferers by making genuine caches in among multiple bluff caches, making it difficult for an observer to accurately trace the caching event [6].

The cache-protection tactics used by jays have recently been compared with techniques employed by magicians to prevent spectators from detecting the cause of their magic effects [13]. For instance, jays manipulate food items within their beaks, akin to the magician's use of 'sleight of hand', to discretely make genuine caches amidst bluff caches. During caching events, jays also conceal items in their throat pouch similar to the magician's use of false pockets [14]. A growing number of studies that use magic as a framework to investigate the human mind have revealed that magicians capitalize on the blind spots in our perception and the roadblocks in our cognition [15–22]. Could jays be susceptible to analogous psychological constraints?

In a recent perspective piece in *Science*, we advocate the application of magic effects to animal psychology as an alternative and innovative avenue for investigating the animal mind [23]. It is worth noting, however, that the use of magic effects has already permeated the methodological paradigms of comparative psychologists. Specifically, researchers have used violation of expectation paradigms on a range of taxa including primates [24,25], dolphins [26], dogs [27] and corvids [28], whereby subjects are presented with both expected and unexpected events. Violations of expectation paradigms are comparable to magic effects because both aim to evoke surprise in the observer by presenting the unexpected. However, in these experiments, researchers primarily rely on looking time as a behavioural measurement, with longer looking times representing 'surprise' when expectations are violated. This single scale measurement makes it difficult to draw cognitive parallels between the human and animal mind when experiencing the unexpected.

Violation of expectation paradigms has also been adapted to include both positive and negative surprises whereby bad food is 'magically' transformed into good food and vice versa [24,29]. Such studies have supplemented looking time with other behavioural measurements such as latency to retrieve the food and likelihood to reject the food. But the current scope of studies has yielded ambiguous results. For example, Tinklepaugh [24] showed that macaques, *Macaca mulatta*, only responded to negative surprises with increased latency to eat the substituted less desirable food. By contrast, Bräuer & Call [29] found that apes (*Pan troglodytes*, *Pan paniscus*, *Pongo pygmaeus*, *Gorilla gorilla*) and dogs, *Canis familiaris*, react to both positive and negative surprises with increased looking and smelling behaviour; but the subjects never rejected the less desirable food and latency to retrieve the food did not differ across expected and unexpected trials [29]. These differing responses could be attributed to the number of trials within each design, which varied across the studies. Specifically, the ape and dog subjects received a limited amount of 'expected' trials before experiencing an 'unexpected' trial, whereas Tinklepaugh's monkeys received hundreds of expected trials before experiencing an unexpected trial. The elevated number of trials in the Tinklepaugh study makes it difficult to determine whether the monkeys were indeed sensitive to the content of the hidden food and thus surprised by the unexpected or whether they were simply responding to the negative contrast effect between the expected and unexpected trials. Negative contrast responses have been demonstrated in a range of species including mammals [30–33], birds [34] and insects [35]. Contrast effects occur when an animal gains information (through reiterate experiences) to create an expectation about a reward and this expectation is not met. For example, evidence in rats suggests that when an expected reward is omitted from a trial they respond with increased aggression [36] and their stress hormones are activated [37].

To gain a better understanding of whether the animal mind is sensitive to cognitive illusions, subjects should be presented with magic effects that would deceive a human spectator [23]. Moreover, cognitive illusions can be amplified if the spectator is given the opportunity to interact with the magician or the props used in the effect [38]. Although the experience of magic is typically a mental phenomenon, it has also been referred to as a social phenomenon because experiencing the impossible can be influenced by the social dynamics between the magician and the members of his audience [39]. Anecdotes from magicians reveal that human spectators exhibit varied responses to magic performances and that this variation might be linked to social dominance [40]. Specifically, spectators that have a higher social dominance rank appear to be more likely to react negatively towards a cognitive illusion. Cache-protection tactics performed by jays can also be described as a social phenomenon because jays will flexibly change their tactics depending on the observer. For instance, jays deprive competitors of visual and acoustic information that might reveal the location of a cache.

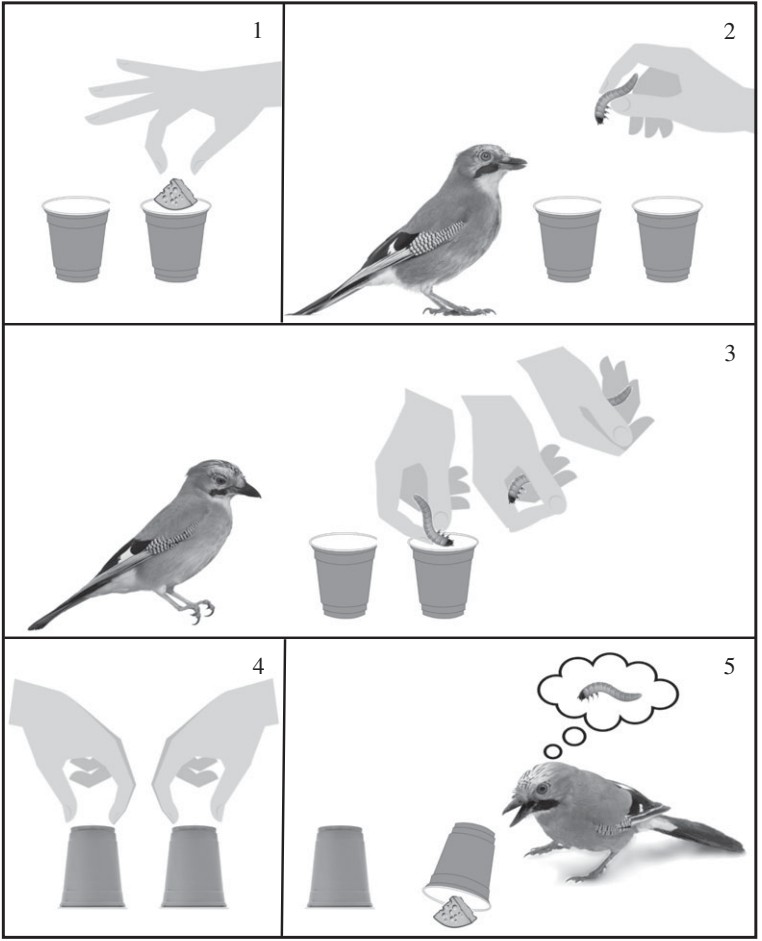

**Figure 1.** The baiting procedure for the de-value condition: (1) out of sight of the bird, a piece of cheese (second preferred food item) is inserted into one of the two cups; (2) both cups are then placed in front of the subject and the bird is presented with the worm (most preferred food item); (3) a dummy-drop is conducted, whereby the worm is inserted into the cup but instead of actually dropping it into the cup, it is concealed within the experimenter's fingers and subsequently withdrawn; (4) both cups are then turned upside down; (5) the bird is allowed to lift either cup using a beak-sized handle to obtain the hidden food item.

By contrast, jays refrain from applying these cache-protection tactics if they are being watched by their mate with whom they will share their cache at a later time [6]. Food-caching jays are, thus, particularly suitable candidates to investigate whether social factors can influence responses to cognitive illusions in non-human minds.

Here, we apply a magical framework, an adapted version of cups-and-balls,[1] to investigate whether observing Eurasian jays, *Garrulus glandarius*, are sensitive to cognitive illusions. The cups-and-balls routine is typically described as a cognitive illusion [41] because it is used to manipulate human perception and cognition by creating an illusion in which balls 'magically' appear and disappear under the cover of opaque cups [42,43]. In humans, the cognitive mechanisms underlying this well-known illusion have been extensively studied by psychologists who have aimed at disentangling the epicentral elements of the experience [41,43,44]. Here, for the first time, we adapt this illusion to investigate responses in non-human animals. The application of this illusion allowed us to not only test whether jays are sensitive to the content of hidden items but also whether their responses are augmented or diminished depending on how the content is changed. We presented jays with two control conditions in which the hidden food was consistent with their expectations and two experimental conditions in which the hidden food was manipulated to violate the birds' expectations. Manipulation conditions, one which elicited a positive surprise and one which elicited a negative surprise (figure 1), were achieved by using a magic-switch, a strategy for creating the perception of a magical effect. We predicted that subjects would react to both positive and negative surprises but

---

[1]Other variations of this magic routine include 'Chop Cup' and the 'Pea & Shell Game'.

R. Soc. Open Sci. **8**: 203358

responses to negative surprises would be stronger. We also measured the social rank of the subjects to determine whether social dominance influenced the jays' responses to cognitive illusions. We predicted that more dominant subjects would exhibit stronger responses to negative surprises.

# 2. Methods

## 2.1. Subjects

We tested the maximum number of jays in our laboratory that would voluntarily participate in the experiment and maximized trial numbers per subject (i.e. 25 trials per condition) [45,46]. Nine sub-adult jays (five years old) from a long-term, stable social group of 16 birds participated in this study from October 2019 to January 2020. However, three jays stopped interacting with the apparatus and only six jays (four females and two males) completed all training and testing. Birds were housed at the Sub-Department of Animal Behaviour, University of Cambridge in Madingley, Cambridgeshire, UK. Housing consisted of an outdoor aviary ($20 \times 10 \times 3$ m) with indoor compartments ($2 \times 1 \times 2$ m) at one end. Birds were able to enter the indoor compartments from the aviary via opaque flap doors ($0.5 \times 0.5$ m), whereby access was controlled by the experimenter. Birds were provided with a maintenance diet (a mixture of soaked dog biscuits, vegetables, eggs, seeds and fruit) in the outside aviary and had *ad libitum* access to water.

## 2.2. Procedure

To ensure that the jays were mildly hungry and thus, motivated to participate in the food-rewarded experiments, the maintenance diet was removed from the aviary 1 h prior to testing. Subjects initiated the daily training and/or testing session on a voluntary basis by entering the open flap doors and moving into the final indoor compartment, which was reserved as a testing chamber. Inside the testing chamber, individual subjects were required to sit on a perch in front of a testing platform that was adjacent to a testing window. The experimenter stood in an adjacent compartment and completed the training and testing via the testing window.

## 2.3. Pre-test training

All subjects were first tested in an object permanence experiment comprised of four tasks [47–49] (electronic supplementary material, table S1), in which performance hinged on understanding that an object continues to exist even if it can no longer be seen. Subjects were then trained to lift an upside-down cup, which was fixed with a piece of string (0.4 cm diameter cotton string) to create a handle that the birds could manipulate using their beaks. Jays were first presented with a worm covered with a clear cup, providing the birds with a clear view of the food item inside. Training with the clear cup ceased once the jays were able to pick up the clear cup using the attached string in 10 out of 10 consecutive trials. Individual subjects were then presented with a worm covered with an opaque red cup, whereby birds were unable to see the food item inside the cup. Jays were trained until individuals had reached the same acquisition criterion used during the clear cup training (i.e. 10 out of 10 consecutive trials).

One week prior to testing, we conducted food preference trials to determine individual preferences in which we identified a preference rank for three commonly consumed food items, including waxworms, peanuts and cubes of cheese ($0.5 \times 0.5 \times 0.5$ cm). Preference trials were tested across three sessions of 12 trials with an inter-test session interval of 24–48 h. In each session, subjects were offered each possible combination of the different food items four times (i.e. waxworms versus peanut; waxworm versus cheese; peanut versus cheese) and each trial was spaced approximately 30 s apart to allow the individual to consume the chosen food item or store the item in its throat pouch. The experimenter presented each combination of the different food items on a raised platform directly in front of the perch approximately 150 mm apart and equidistant to the bird. The position of the food item (located to the left or the right of the subject) was randomly switched but counter-balanced across each session. The item that the subject initially approached was given to the bird and rated as the preferred food item and the alternative food item was immediately removed by the experimenter. Individual food preferences differed across subjects (table 1). The most preferred and second preferred food items pertaining to each individual were subsequently used as their rewards during the magic-switch experiment.

**Table 1.** Food preferences for each subject.

| subject | most preferred | second preferred |
|---|---|---|
| Chinook | peanut | waxworm |
| Dolci | waxworm | cheese |
| Homer | peanut | waxworm |
| Jaylo | waxworm | cheese |
| Poe | waxworm | cheese |
| Stuka | peanut | waxworm |

## 2.4. Magic-switch experiment

This experiment was inspired by the well-known magic routine cups-and-balls.[2] Traditionally, the routine involves three empty cups and three small balls with countless adaptations. Typically, a magician makes the balls appear in random cups, disappear entirely or 'magically' transforms them into unexpected items. In our adapted version of cups-and-balls, we used food items instead of balls and we used two cups fitted with a piece of a string so that the birds could lift them using their beaks. Specifically, jays were presented with two identical up-turned red cups on a raised platform directly in front of the perch approximately 150 mm apart and equidistant to the bird. They were first presented with a baiting process in one of the two cups and then the experimenter flipped both cups upside-down. Subjects were subsequently able to uncover either cup to retrieve the hidden food reward (figure 1).

Subjects were presented with four different baiting conditions including two control conditions in which the hidden food item was consistent with the birds' expectations and two experimental conditions in which the hidden food item was manipulated to violate the birds' expectations. In control 1, the subject's most preferred food item was baited in full view and the same food item subsequently appeared when the subject uncovered the cup. In control 2, the subject's second preferred food item was baited in full view and the same food item subsequently appeared when the subject uncovered the cup. In the up-value condition, the subject's second preferred food item was bluff baited, but the most preferred food item subsequently appeared when the subject uncovered the cup. This condition was conducted to simulate a positive surprise. In the de-value condition, the subject's most preferred food item was bluff baited, but the second preferred food item subsequently appeared when the subject uncovered the cup. This condition was conducted to simulate a negative surprise (figure 1).

Manipulations in the experimental trials were achieved by placing an unexpected food item inside one of the two cups out of sight of the bird. The two cups were subsequently placed in front of the subject and the bird was presented with the expected food item in the experimenter's hand. The experimenter then conducted a dummy-drop, whereby the expected food item was inserted into the cup but instead of actually dropping it into the cup, it was concealed within the experimenter's fingers and subsequently withdrawn (figure 1). To ensure that the dummy-drop technique was effectively deceiving the birds, we also conducted a subset of the control trials using the same procedure. Specifically, out of sight of the bird, the expected food item was placed inside one of the two cups, and then in full view of the bird, the experimenter used a dummy-drop to bluff bait the same cup with the expected food item. We found no significant differences in the birds' responses between the standard control trials and the subset of control trials that were completed using the dummy-drop technique, and thus these data were pooled.

Testing sessions were completed across a period of 14 days. Subjects received between eight and 10 testing sessions comprised of up to 12 trials. Each session consisted of three trials of each of the four conditions (i.e. control 1, control 2, up-value and de-value) presented in pseudo-randomized order to ensure that they did not receive more than two trials of the same condition in successive order. This was conducted to exclude the influence of negative and positive contrast effects. As participation was voluntary, some subjects left the testing compartment prior to completing all 12 trials. When this occurred, testing would then resume on the following day. For example, if a subject completed six trials in a single session, the following day they would start the session on the seventh trial. In order to prevent the subjects from developing a side bias, the placement of the food item was pseudo-randomized across both cups but counter-balanced across each session so that the food item was

[2]Other variations of this magic routine include 'Chop Cup' and the 'Pea & Shell Game'.

**Table 2.** Dominance rank of our sample ($N = 6$ Eurasian jays).

| subject | dominant | subordinate | degree of dominance (%) |
|---|---|---|---|
| 1. Homer | Stuka, Dolci and Chinook | — | 50 |
| 2. Dolci | Stuka, Poe and Jaylo | Homer and Chinook | 17 |
| 3. Stuka | Jaylo and Chinook | Dolci and Homer | 0 |
| 4. Poe | Chinook | Dolci | 0 |
| 5. Chinook | Dolci | Stuka and Poe | −17 |
| 6. Jaylo | — | Stuka and Dolci | −33 |

never inserted more than twice in a row into the same cup. Given that we were limited by resource constraints, we maximized the number of trials per condition for each subject to help maintain statistical power [45]. However, trials per condition did not exceed 25 since over-exposure to the cups-and-balls routine in humans can improve perceptual performance [41]. Testing sessions were recorded using a GoPro® Hero 6 video-camera and later analysed.

## 2.5. Scoring social rank

We determined the social rank of each subject within our sample by conducting a series of observational feeding sessions. The sessions were carried out for 15 min every weekday for four consecutive weeks. Their daily maintenance diet was placed in an elevated feeder where the subject sample could feed and interact for 15 uninterrupted minutes while being recorded. Social interactions were recorded using the same video camera used for the testing procedure and later analysed. Dominant behaviours were defined as aggressive actions that were exhibited by one subject to another including chasing the other bird away from the feeding platform, beak snapping and attacking. For a subject to be considered dominant over another subject, the bird would have to exhibit a minimum of four dominant behaviours towards the subordinate bird. Once all feeding sessions were analysed, we calculated an overall degree of dominance percentage (*DDP*) for each subject using the following formula:

$$DDP = \left(\frac{Nd - Ns}{S}\right) 100\%,$$

where *Nd* represents the total amount of dominance, *Ns* represents the total amount of subordinance and *S* represents the subject sample. For example, our analysis of the feeding sessions demonstrated that Dolci was dominant over three subjects (Stuka, Poe and Jaylo) and subordinate to two subjects (Homer and Chinook) (*DDP* = (3 − 2/6) × 100%) giving her an overall dominance score of 17% (table 2).

## 2.6. Data scoring and statistical analyses

All trials were analysed using the video recordings. We coded behaviours for a 30 s period for each trial because trials were spaced approximately 30 s apart. The start of this period commenced the moment the subject lifted the cup to uncover the food reward and ended after 30 s. We later reduced the scoring period to 10 s because all subjects would either approach and consume the uncovered food item within 10 s or outright reject the reward within this time period. Rejection occurred when the subject would uncover the food reward but instead of retrieving it, the bird would leave the testing perch and platform only to return once the rejected food reward was removed and the experimenter motioned to the subject that a new trial was about to commence. We monitored specific behavioural measurements including latency to consume and checking behaviour. We coded the latency to consume the uncovered food reward by recording the number of seconds that elapsed from the moment the subject lifted the baited cup to the moment when the subject retrieved the food reward. Instances when the subject rejected the food reward were assigned a maximum latency of 10 s. We also coded whether subjects re-checked the baited cup after the initial uncovering by either lifting it again or looking inside the cup. Moreover, we coded whether subjects checked the alternative cup after uncovering the baited cup.

Statistical analyses were completed using JASP (v. 0.10.2) and RStudio for Mac (v. 1.2.1335). To assess inter-rater reliability, another experimenter scored a pseudo-random selected sample of 20% of the trials ($N = 120$),

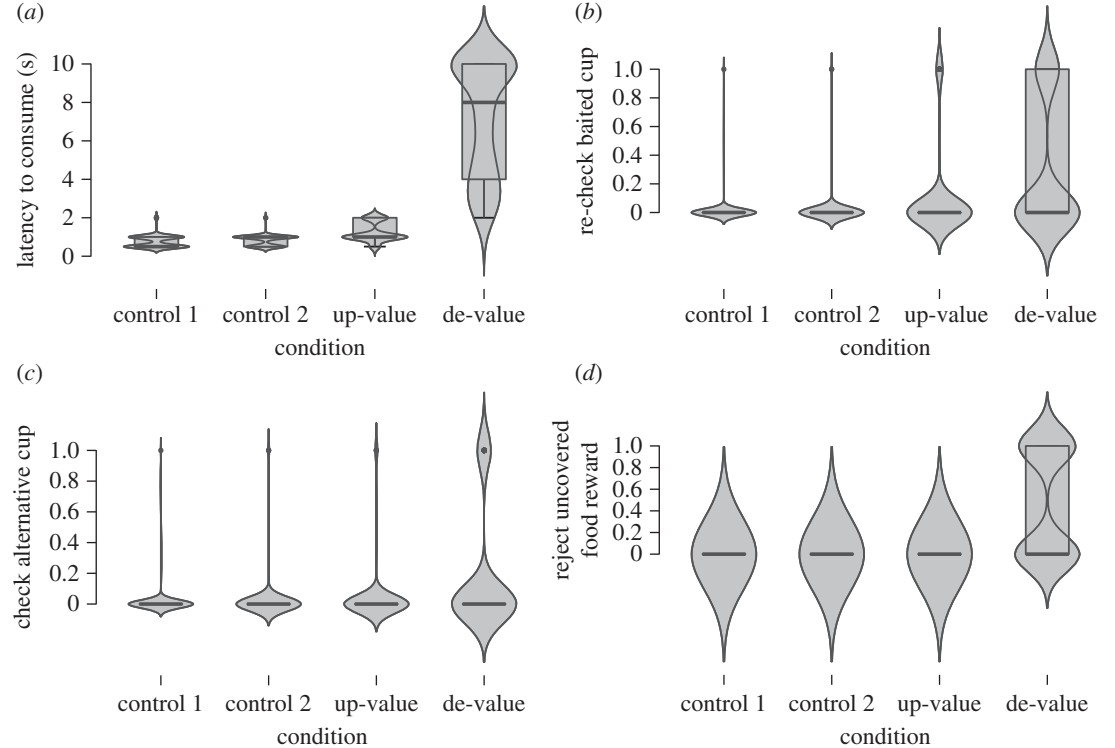

**Figure 2.** Responses to the four baiting conditions, $N = 6$ Eurasian jays. Boxplots depict (a) latencies to consume food item; (b) the proportion of instances when subjects re-checked the baited cup; (c) the proportion of instances when subjects checked the alternative cup and (d) the proportion of rejection behaviour after uncovering the baited cup.

which included a balanced number of the four conditions. Reliability was excellent for latency to consume (Cohen's Kappa = 0.87), for re-checking the baited cup (Cohen's Kappa: 0.90), for checking the alternative cup (Cohen's Kappa: 0.95), and for rejection of the uncovered food reward (Cohen's Kappa = 0.96).

To investigate the influence of condition on the subjects' latency to consume the food reward, checking behaviour and reject behaviour, we conducted repeated-measures analyses. First, we checked parametric assumptions using Shapiro–Wilks tests. Results revealed that our data were not normal (latency: $W = 0.620$, $p < 0.001$; re-check: $W = 0.332$, $p < 0.001$; check other: $W = 0.247$, $p < 0.001$; reject: $W = 0.383$, $p < 0.001$) (electronic supplementary material). We thus proceeded with non-parametric permutation tests (aovperm function, permuco package). Significant differences were further explored using pairwise permutation $t$ tests with a Holm–Bonferroni adjustment to maintain the overall alpha level at the nominated value of 0.05 for multiple pairwise comparisons (pairwise.perm.t.test function, RVAideMemorie package). To investigate whether rejection behaviour was correlated with social rank, we used a two-tailed Kendall's tau correlation coefficient, which is appropriate for small samples (less than 25) [50]. The variables in the correlation analysis included rejection rate and the degree of dominance. Given that we had a limited sample size of six, we also examined these data to determine the strength of the evidence in support of the alternative or the null hypothesis. To do this, we calculated a Bayes factor using Bayesian Information Criteria [51], comparing the fit of the data under the null and the alternative hypothesis ($BF_{10}$ = alternative/null).

# 3. Results

## 3.1. Latency to consume food

We measured the subjects' latency to consume the food reward once they uncovered the red cup. Latency to consume the food reward was significantly different across conditions ($F_3 = 117.50$; $p < 0.001$). The de-value condition that simulated a negative surprise elicited the strongest response in the jays, with individuals exhibiting longer latencies to consume the uncovered food item compared with all other conditions (all $p_{holm} < 0.001$) (figure 2a). Latencies were also significantly different between the up-value condition and both control conditions (both $p_{holm} < 0.001$). In response to both controls, when

the food items were consistent with the birds' expectations, subjects readily accepted the hidden food item; and latencies to consume the most preferred food and second preferred food were comparable ($p_{holm} = 0.085$) (figure 2a).

## 3.2. Likelihood of re-checking the baited cup

We measured the subjects' likelihood to re-check the cup after the initial uncovering. Re-checking was significantly different across conditions ($F_3 = 91.98$; $p < 0.001$). The jays were more likely to re-check the baited cup after the initial uncovering in response to the de-value condition compared with both control conditions (both $p_{holm} < 0.001$) and compared with the up-value condition ($p_{holm} < 0.01$) (figure 2b). Similarly, the jays were significantly more likely to re-check the baited cup in response to the up-value condition compared with both controls (both $p_{holm} < 0.01$). By contrast, the jays scarcely re-checked the baited cup in response to control 1 and control 2 and re-checking behaviour was comparable across both controls ($p_{holm} = 0.994$) (figure 2b).

## 3.3. Likelihood of checking the alternative cup

We measured the subjects' likelihood to check under the alternative cup once they had uncovered the baited cup. The subjects' likelihood to check the alternative cup was significantly different across the conditions ($F_3 = 18.96$; $p < 0.001$). The jays were more likely to check the alternative cup in response to the de-value condition compared with both control conditions (both $p_{holm} < 0.001$) and compared with the up-value condition ($p_{holm} < 0.01$) (figure 2c). Checking the alternative cup was rarely observed in response to control 1, control 2 and the up-value condition, and this behaviour was comparable across these three conditions (control 1–control 2: $p_{holm} = 1.00$; control 1–up-value: $p_{holm} = 0.65$; control 2–up-value: $p_{holm} = 1.00$) (figure 2c).

## 3.4. Rejection rate

To further investigate the discrepancy between the responses evoked by the positive and negative surprise, we also measured whether any of the conditions in the adapted cups-and-balls routine evoked rejection behaviour in the jays, whereby the subject would refuse to take the food item once they uncovered the baited cup. Rejection was significantly different across the conditions ($F_3 = 52.49$; $p < 0.001$). We found that the jays were significantly more likely to reject the food item in response to the de-value condition (average rejection = 46.67%) compared with all other conditions (all $p_{holm} < 0.001$). Moreover, rejection of the food item did not occur in any of the other conditions (all $p_{holm} = 1.00$) (figure 2d).

## 3.5. Rejection rate and social rank

We found that subjects that exhibited a higher rejection rate tend to be more dominant. Specifically, rejection rate in response to the de-value condition was significantly correlated with the degree of dominance within our sample (Kendall's tau $B = 0.828$, $p < 0.05$) (figure 3). This suggests that less dominant jays were more likely to accept the de-valued food item, whereas more dominant jays were more likely to reject the de-valued food item. To determine the strength of the evidence in favour of there being a correlation, we also analysed the relationship between rejection behaviour and social rank using a Bayesian correlation matrix. This analysis demonstrated that the strength of evidence is moderate ($BF_{10} = 4.178$). The estimated Bayes factor indicates that our data were 4.178 times more likely to be observed under the alternative hypothesis (i.e. significant correlation) than the null hypothesis (i.e. no correlation).

# 4. Discussion

Using an adapted cups-and-balls routine typically performed to humans, we demonstrate that jays are sensitive to cognitive illusions. The response pattern to the different conditions in the adapted cups-and-balls routine demonstrates that the birds were only sensitive to the value of the different food items in the manipulation conditions when the hidden items violated their expectations. This suggests that our subjects' responses cannot solely be explained by their food preferences and that the jays are

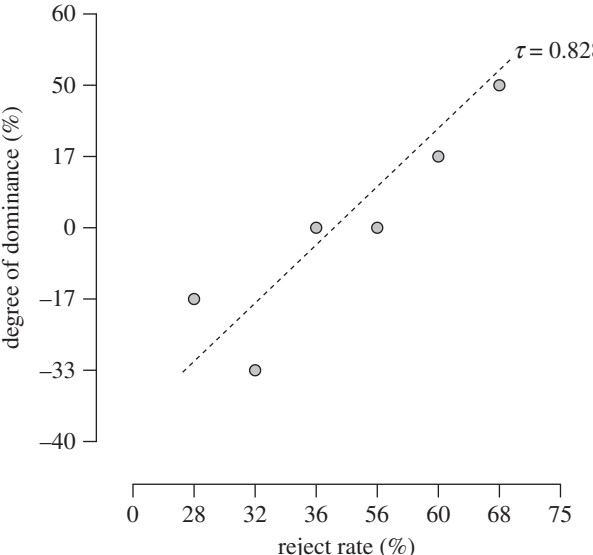

**Figure 3.** Rejection behaviour and social rank in $N = 6$ Eurasian jays. Relationship between the reject rate in response to the de-value condition ($N$ trials = 24 per subject) and the degree of dominance within our sample. The relationship is indicated by a two-tailed Kendall's tau correlation coefficient. Correlation analyses between the other conditions and social rank were not conducted because rejection of the reward was only observed in the de-value condition, whereby the most preferred food item was magically switched for the second preferred food item; $*p < 0.05$.

indeed sensitive to cognitive illusions. Within the context of caching, previous research has shown that jays are sensitive to the content of their own caches and will discriminately retrieve items depending on their food preferences and the perishability of the cached item [52,53]. Our study suggests that, as observers, jays might impose the same content sensitivity when they pilfer the caches of others.

As predicted, the jays responded to both positive and negative surprises, which is evident from the differences in latency and re-checking behaviour between the control and manipulated conditions, but the jays were also sensitive to the type of cognitive illusion. Specifically, the jays tolerated the cognitive illusion when it led to a more advantageous scenario (i.e. when the reward was more valuable than expected) but rejected the hidden food items when they were less valuable than expected (electronic supplementary material, video S1). This pattern of behaviour suggests that jays, like some primates, respond asymmetrically to positive and negative events [54–57]. For example, monkeys respond negatively when they expect an item of higher value (i.e. piece of banana) but receive a less-valued food item (i.e. piece of lettuce) [24]. In humans, negative events tend to receive more attention and can prompt increased cognitive processing compared with positive events [58–61]. This disparity has been investigated within the context of gains and losses and previous research has demonstrated that humans can experience stronger emotional responses towards a loss as opposed to a gain [62], a pattern consistent with loss aversion [63,64].

To put this into context, imagine you are approached by a magician who allures you with the promise of gifting you the currency note that he holds in his hand. First, the magician holds out a £5 note for your inspection and then proceeds to fold the note again and again until it resembles a tightly folded package. The magician presents you with the package and asks you to unfold the note. You unfold the package to reveal that the £5 note has 'magically' transformed into a £20 note, a positive surprise that generates a sense of wonder and elation. Now consider that the magician had created the reverse effect and 'magically' transformed a £20 note into a £5 note, a negative surprise that might generate feelings of dissatisfaction and frustration. In both scenarios, the cognitive illusion elicits a violation of expectation, and the spectator uses the same amount of cognitive and physical effort to observe the magic effect and unfold the note, respectively. However, the surprise aspect might vary in strength and evoke considerably different responses depending on whether the cognitive illusion simulates a loss or a gain.

Similar cognitive processes might be occurring in the jays. Indeed, despite using the same amount of cognitive effort to track the hidden food item in the adapted cups-and-balls routine, the influence of the jays' losses (i.e. de-value) was greater than their gains (i.e. up-value). This is reflected in the increased likelihood to re-check the baited cup and check the alternative cup in response to the de-value condition, as well as the significantly higher likelihood to reject the food item in the de-value

condition. However, this study alone cannot offer conclusive evidence that the magic-switch experiment is exploiting similar cognitive mechanisms in jay observers to those that might be exploited in human observers. Notice that we use an adapted version of the cups-and-balls effect and thus the experience of the jays is likely going to differ from that of how humans might experience the traditional cups-and-balls effect. Further research with comparative analyses are required to pinpoint the similarities and differences between human and non-human observers when perceiving cognitive illusions. Indeed, one recent study suggests that Eurasian jays have different expectations to humans when observing cognitive illusions using sleight of hand techniques [65]. The study found that, similar to humans, jays were deceived by cognitive illusions that used fast movements as a deceptive action. By contrast, unlike humans, jays were not deceived by cognitive illusions that capitalized on the observer's expectations with respect to how human hands might manipulate hidden objects [65].

Consistent with our hypothesis, rejection was more prevalent in socially dominant subjects. Despite our small sample size, the Bayes factor is indicative of at least moderate evidence supporting a correlation between rejection rate and social rank and thus this area of research deserves further attention. Nevertheless, consider the reported correlation statistics with caution since our limited sample size could lead to an underpowered correlation analysis. Replications of the current study with a larger sample size might highlight finer-grained analyses and could improve the evidential value of our findings. Patterns between reward rejection and social rank might have ecological relevance because less dominant individuals, who have less access to food resources, need to accept less favourable food [66]. By contrast, more dominant individuals, who have greater access to optimal food resources, can afford to be more discerning [67–69].

Taken together, our study demonstrates that observer jays remember the content of hidden food items and this sensitivity is amplified if the cognitive illusion simulates a loss. This pattern suggests that pilfering jays might be using a combination of cognitive abilities such as memory, imagining the future and evaluating expectations to guide their decisions about whether to steal the caches of others. Future research should focus on whether caching jays exploit the psychological constraints of observing pilferers in order to evoke deceptive tactics involved in cache-protection. The application of magical frameworks to comparative cognition unlocks alternative and innovative avenues for investigating the animal mind. The focus of comparative cognition studies is usually centred on investigating the evolution of analogous cognitive aptitude across species. However, the use of magic effects provides researchers with an opportunity to look at the other side of the coin—the evolution of analogous psychological constraints in diverse animal minds.

Ethics. The experiments were approved by the University of Cambridge and conducted under a non-regulated procedure (no. zoo72/19).

Data accessibility. The data are provided in electronic supplementary material [70].

Authors' contributions. All authors contributed to the conceptualization of the study with N.S.C. as PI. A.K.S. designed the experimental paradigm. A.K.S. and N.C.S. validated the experimental methodology. A.K.S., M.L. and E.G.P. collected the data. A.K.S. analysed the data. All authors participated in the interpretation of the data. A.K.S. drafted the manuscript and all authors edited and approved the final version. N.S.C. directed and supervised the project.

Competing interests. We have no competing interests, and all authors declare that the research was conducted in the absence of any commercial or financial relationships that could be construed as a potential conflict of interest.

Funding. This work was supported by multiple funding bodies during conception, data collection and writing of this study. A.K.S. was supported by a Newton International Fellowship funded by the Royal Society (NIF\R1\180962). M.L. was supported by a PhD scholarship funded by the CA.RI.PA.RO. Foundation. C.W. was supported by an Artist Residence Program in the Department of Psychology. N.S.C. was supported by a European Research Council (FP7/2007–2013)/ERC grant agreement no. 3399933.

Acknowledgements. Many thanks to the funding bodies that supported the authors. We thank staff at the Sub-Department of Animal Behaviour for assistance with animal husbandry. We also offer many thanks to all members of the Comparative Cognition Laboratory for helpful discussions about magic and cognition.

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
