## [Peer Review File · Royal Society Open Science]

Review History

RSOS-202358.R0 (Original submission)

Review form: Reviewer 1

Is the manuscript scientifically sound in its present form?

Yes

Are the interpretations and conclusions justified by the results?

Yes

Is the language acceptable?

Yes

Do you have any ethical concerns with this paper?

No

Have you any concerns about statistical analyses in this paper?

No

Recommendation?

Accept with minor revision (please list in comments)

Comments to the Author(s)

RSOS-202358

Authors: Schnell, Loconsole, Garcia-Pelegrin, Wilkins, & Clayton.

A frequent question in comparative cognition is whether two different species of animals are able to successfully address the same environmental challenge. Here the question is inverted: Are humans and Eurasian jay's subject to the same misperceptions (i.e., illusions). Specifically, the manuscript reports a study of Eurasian jays' sensitivity to a slight-of-hand that creates a discrepancy between prior reward experience and current reward for the same response. This parallels humans misperceiving the location of a reward due to slight-of-hand. Observer jays were found to reinspect a receptacle more often when the reward was altered relative to prior experience with that receptacle, and they were slower to consume the food reward when the reward was devalued relative to prior experience. In fact, more dominant jays often completely rejected the devalued reward when their 'expectations' were higher. The report adds to the many illuminating findings concerning avian cognition from Clayton's lab in the past. The descriptions of what was done and what was observed are clear and appropriate.

My one significant concern is that the research was designed to determine whether the jays are subject to the same visual illusion produced by slight-of-hand that humans are. There is a jump from the behavior of the birds to the conclusion that they are subject to the same illusion as humans are. I think that the wording needs to be modified a bit to better differentiate between the observed behavior and the conclusion that the underlying illusion is the same as humans appear subject to. The conclusion at the cognitive level that an illusion was experienced is reasonable, but rests on an assumption concerning the subjects' reasoning to arrive at their behavior. I am not suggesting that this assumption is incorrect, only that there is an assumption here that really goes beyond the data and should be identified as such.

Of lesser concern:

It is surprising to me that there was so little effort was made to connect the present observations with the extensive literature on successive contrast effect, which ordinarily yields results highly consistent with the present observations. In these earlier reports, devaluation led to reduced effort [a so-called depression effect] akin to the slower consumption observed here.

The abstract fails to adequately focus on what was observed and the implications of what was observed in the present study. The first three sentences of the abstract seem superfluous.

I wish that "Furthermore, if they are being observed by a competitor, jays will adjust their cache-protection behaviour in response to whether the observer is subordinate or dominant (6)." had been worded such as to tell the reader specifically what had been observed.

Line 3 of Table 2 has an error in it. 'Stuka was dominant to Stuka' makes no sense.

Maybe I just failed to see them, but I think that the authors have failed to provide links to their raw data, data analysis, and procedural details.

Review form: Reviewer 2**Is the manuscript scientifically sound in its present form?**

Yes

Are the interpretations and conclusions justified by the results?

No

Is the language acceptable?

Yes

Do you have any ethical concerns with this paper?

No

Have you any concerns about statistical analyses in this paper?

Yes

Recommendation?

Major revision is needed (please make suggestions in comments)

Comments to the Author(s)

This manuscript reports a study in which jays are presented with a task involving deception: a food item is placed inside a cup, but sometimes it is replaced by a different item (of larger or smaller value) without the bird's awareness. The finding is that jays produce longer latencies and larger rejection rates for food items that are replacing one of larger value than in the rest of conditions.

In general, I think that the research question is interesting, and the study was well conducted. The manuscript describes all the necessary information in a direct and accessible language.

However, I have several concerns, specifically with the statistical analysis and data reporting, which I list below:

-My first comment is conceptual rather than methodological. I am not sure that "cognitive illusion" is the right term to describe what happens in this study. The paper reports a procedure in which expectations are violated by means of a trick. Although I can agree that most (or all) cognitive illusions involve expectation violation to some extent, I do not think that all paradigms that involve violation of expectations can be considered "a cognitive illusion". Certainly, I am not sure that I would use this word if I were observing a similar procedure to the one in this experiment, but in humans. The Introduction section would improve if additional detail is given as to how this study connects with the typical scenarios in which we talk about "cognitive illusions" (in humans).

I do acknowledge, however, a subtle but reasonably well explained connection to the concept of "cognitive bias", as jays seem to fall prey of the well-known "loss aversion" bias.

-The sample size is small (initial N=9, final N=6). This can be fine in many situations, specially in procedures like these that involve nonhuman animals and multiple measures. However, a small N definitely limits the conclusions one can draw from data. In line with current practices to increase transparency, I would like to read in the paper a justification on how the sample size was decided. Ideally, we would have (1) a target effect size from previous studies, or (2) a sensitivity analysis that ensures we can observe meaningful effects with reasonable power (i.e., you can fix N and power, and report the smallest effect size you can detect).

-Also to improve transparency, I would recommend to include all test statistic values and the rest of information necessary to interpret them (e.g., degrees of freedom), as in many analyses we can only see the p-values and effect sizes. Additionally, the effect sizes are not completely reported, as no information about the scale or units is given: e.g., do these values correspond to raw data units, or to standardized measures such as Cohen's d?

-On a similar note, it is recommended to report exact p-values, instead of the old phrase “ $p < 0.05$ ”.

-But the most worrying issues concern the dominance analysis in Figure 3. First, when looking at the figure, it makes the reader wonder whether the analysis is correct. Specifically, with a N of 6 (as this analysis does not contain multiple measures), is it right to compute a Pearson’s r ? Is this estimation reliable? I think that, unless more information is made available, the answer is that the study lacks power for testing this hypothesis.

For illustration, I have computed the minimum effect size that can be observed with reasonable power (0.80) with a $N=6$, and it is $\rho = 0.84$ (i.e., very large). That is, you cannot observe reliably effects smaller than that with $N=6$. Or, put it another way, your power to detect a large effect ($r>0.50$) with that sample size is only 0.19 (i.e., if you repeat your study with $N=6$ 100 times, only 19 times you will get a significant p-value). The power for detecting a medium effect ($r>0.30$) is much smaller (0.09).

What I mean is that the study was not designed to test this hypothesis properly, as it is incapable of detecting medium-to-large effects, not to mention smaller ones. This comment holds unless the authors can justify somehow that the expected effect size is >0.84 .

-Even if we assume that the study was well powered to detect the target effect, the reported statistics seem to be wrong, or at least we lack the information to properly reconstruct them. The authors state: “rejection rate in response to the de-value condition was significantly correlated with the degree of dominance within our sample (Pearson’s $r = 0.791$, $p < 0.05$)”.

Note that no exact p-value is provided in the text. However, it is possible to obtain this information from the N and r values provided. The problem is that, for this coefficient value (0.791) and N (6), the corresponding p-value is 0.061 (assuming a two-tailed test). That is, the results is not significant, contrary to what the paper states. So, if the test was conducted as two-tailed (as is normally by default), then the conclusions are not supported by the data.

-In any case, the authors are providing also a Bayesian analysis to complement the p-value. I appreciate this as I think it can provide useful information. However, I was puzzled as the BF for this analysis is 3.97, which suggests “moderate” evidence for the experimental hypothesis. One of the nice features of the Bayesian approach is that, when information is scarce, the BF produces values pointing to inconclusive results, rather than the false-alarms delivered by p-values. This means that BFs use to be more conservative (with few data) than are p-values. This is what makes the results a bit surprising: with a very small N, we have a nonetheless “high” Bayes Factor, where I would have expected a smaller value.

-I wanted to dig into this issue, but when I downloaded the “raw” dataset provided as supplementary material, I found that the information concerning the social dominance is missing. I think this variable should be part of the shared dataset.

-Then, I reconstructed myself the data from Figure 3. With my reconstructed dataset, $r = 0.792$ (identical to the reported value). The p-value I obtain is 0.061, inconsistent with the results reported. The BF I obtain is also different, $BF=2.67$, below the cut-off point (a value < 3 is considered as “anecdotal evidence for the hypothesis”, equivalent to a nonsignificant p-value). What can explain this discrepancy? I have some possibilities in mind:

First, I thought that the authors were reporting one-tailed tests instead of the standard two-tailed tests, but the numbers still do not coincide.

Second, given the small N, it is possible that they are using a non-parametric analysis that produces a smaller p-value. But then they should clearly explain it in the text (currently, they say “To investigate whether rejection behaviour was correlated with social rank, we used a Pearson’s correlation coefficient”).

Third, the discrepancy with the Bayes factor can be due to a different choice of priors. However, this does not seem to be case as I also conducted a robustness check on my reconstructed data, and no choice of prior in JASP can increase the BF above 2.68.

In sum, after reading the paper and conducting these analyses, I do not know how the results reported in Figure 3 were obtained.

In sum, I think that this is an interesting paper and that the experiment was well conducted. My suggestions to improve the paper are as follows:

-The Introduction needs a better elaboration on how this research is connected to cognitive illusions in humans. I think that the mere reporting of jays using their memory abilities, expectations, and what looks like re-evaluation processes is highly interesting on its own without the reference to illusions.

-The results concerning the dominance variable should be corrected or removed (if they are actually wrong), or completed (if they are right but lack information). With the information that is available, it seems clear that the study was not designed to test this hypothesis with reasonable power.

-The paper needs to incorporate several measures to increase transparency in results reporting. At least: sample size justification, more detail in statistics and how they were computed, and complete dataset available for download.

Decision letter (RSOS-202358.R0)

Dear Professor Schnell

The Editors assigned to your paper RSOS-202358 "Jays are sensitive to cognitive illusions" have now received comments from reviewers and would like you to revise the paper in accordance with the reviewer comments and any comments from the Editors. Please note this decision does not guarantee eventual acceptance.

Please submit your revised manuscript and required files (see below) no later than 21 days from today's (ie 06-May-2021) date. Note: the ScholarOne system will 'lock' if submission of the revision is attempted 21 or more days after the deadline. If you do not think you will be able to meet this deadline please contact the editorial office immediately.

Please note article processing charges apply to papers accepted for publication in Royal Society Open Science (<https://royalsocietypublishing.org/rsos/charges>). Charges will also apply to

papers transferred to the journal from other Royal Society Publishing journals, as well as papers submitted as part of our collaboration with the Royal Society of Chemistry (<https://royalsocietypublishing.org/rsos/chemistry>). Fee waivers are available but must be requested when you submit your revision (<https://royalsocietypublishing.org/rsos/waivers>).

on behalf of Dr Shinya Yamamoto (Associate Editor) and Essi Viding (Subject Editor)
openscience@royalsociety.org

Associate Editor Comments to Author (Dr Shinya Yamamoto):

Comments to the Author:

We apologize for taking time to give you back our decision. It took time to find appropriate reviewers and associate editors have changed in the process. Finally we have two reviewers' comments. They evaluated your manuscript high, but also recommend considerable revision. I hope you can manage to revise it accordingly.

Reviewer comments to Author:

Reviewer: 1

Comments to the Author(s)

RSOS-202358

Authors: Schnell, Loconsole, Garcia-Pelegrin, Wilkins, & Clayton.

A frequent question in comparative cognition is whether two different species of animals are able to successfully address the same environmental challenge. Here the question is inverted: Are humans and Eurasian jays subject to the same misperceptions (i.e., illusions). Specifically, the manuscript reports a study of Eurasian jays' sensitivity to a slight-of-hand that creates a discrepancy between prior reward experience and current reward for the same response. This parallels humans misperceiving the location of a reward due to slight-of-hand. Observer jays were found to reinspect a receptacle more often when the reward was altered relative to prior experience with that receptacle, and they were slower to consume the food reward when the reward was devalued relative to prior experience. In fact, more dominant jays often completely rejected the devalued reward when their 'expectations' were higher. The report adds to the many illuminating findings concerning avian cognition from Clayton's lab in the past. The descriptions of what was done and what was observed are clear and appropriate.

My one significant concern is that the research was designed to determine whether the jays are subject to the same visual illusion produced by slight-of-hand that humans are. There is a jump from the behavior of the birds to the conclusion that they are subject to the same illusion as humans are. I think that the wording needs to be modified a bit to better differentiate between the observed behavior and the conclusion that the underlying illusion is the same as humans appear subject to. The conclusion at the cognitive level that an illusion was experienced is reasonable, but rests on an assumption concerning the subjects' reasoning to arrive at their behavior. I am not suggesting that this assumption is incorrect, only that there is an assumption here that really goes beyond the data and should be identified as such.

Of lesser concern:

It is surprising to me that there was so little effort was made to connect the present observations with the extensive literature on successive contrast effect, which ordinarily yields results highly consistent with the present observations. In these earlier reports, devaluation led to reduced effort [a so-called depression effect] akin to the slower consumption observed here.

The abstract fails to adequately focus on what was observed and the implications of what was observed in the present study. The first three sentences of the abstract seem superfluous.

I wish that "Furthermore, if they are being observed by a competitor, jays will adjust their cache-protection behaviour in response to whether the observer is subordinate or dominant (6)." had been worded such as to tell the reader specifically what had been observed.

Line 3 of Table 2 has an error in it. 'Stuka was dominant to Stuka' makes no sense.

Maybe I just failed to see them, but I think that the authors have failed to provide links to their raw data, data analysis, and procedural details.

Reviewer: 2

Comments to the Author(s)

This manuscript reports a study in which jays are presented with a task involving deception: a food item is placed inside a cup, but sometimes it is replaced by a different item (of larger or smaller value) without the bird's awareness. The finding is that jays produce longer latencies and larger rejection rates for food items that are replacing one of larger value than in the rest of conditions.

In general, I think that the research question is interesting, and the study was well conducted. The manuscript describes all the necessary information in a direct and accessible language.

However, I have several concerns, specifically with the statistical analysis and data reporting, which I list below:

-My first comment is conceptual rather than methodological. I am not sure that "cognitive illusion" is the right term to describe what happens in this study. The paper reports a procedure in which expectations are violated by means of a trick. Although I can agree that most (or all) cognitive illusions involve expectation violation to some extent, I do not think that all paradigms that involve violation of expectations can be considered "a cognitive illusion". Certainly, I am not sure that I would use this word if I were observing a similar procedure to the one in this experiment, but in humans. The Introduction section would improve if additional detail is given as to how this study connects with the typical scenarios in which we talk about "cognitive illusions" (in humans).

I do acknowledge, however, a subtle but reasonably well explained connection to the concept of "cognitive bias", as jays seem to fall prey of the well-known "loss aversion" bias.

-The sample size is small (initial N=9, final N=6). This can be fine in many situations, specially in procedures like these that involve nonhuman animals and multiple measures. However, a small N definitely limits the conclusions one can draw from data. In line with current practices to increase transparency, I would like to read in the paper a justification on how the sample size was decided. Ideally, we would have (1) a target effect size from previous studies, or (2) a sensitivity analysis that ensures we can observe meaningful effects with reasonable power (i.e., you can fix N and power, and report the smallest effect size you can detect).

-Also to improve transparency, I would recommend to include all test statistic values and the rest of information necessary to interpret them (e.g., degrees of freedom), as in many analyses we can only see the p-values and effect sizes. Additionally, the effect sizes are not completely reported, as no information about the scale or units is given: e.g., do these values correspond to raw data units, or to standardized measures such as Cohen's d?

-On a similar note, it is recommended to report exact p-values, instead of the old phrase "p < 0.05".

-But the most worrying issues concern the dominance analysis in Figure 3. First, when looking at the figure, it makes the reader wonder whether the analysis is correct. Specifically, with a N of 6 (as this analysis does not contain multiple measures), is it right to compute a Pearson's r? Is this estimation reliable? I think that, unless more information is made available, the answer is that the study lacks power for testing this hypothesis.

For illustration, I have computed the minimum effect size that can be observed with reasonable power (0.80) with a N=6, and it is $\rho = 0.84$ (i.e., very large). That is, you cannot observe reliably effects smaller than that with N=6. Or, put it another way, your power to detect a large effect ($r > 0.50$) with that sample size is only 0.19 (i.e., if you repeat your study with N=6 100 times, only 19 times you will get a significant p-value). The power for detecting a medium effect ($r > 0.30$) is much smaller (0.09).

What I mean is that the study was not designed to test this hypothesis properly, as it is incapable of detecting medium-to-large effects, not to mention smaller ones. This comment holds unless the authors can justify somehow that the expected effect size is > 0.84 .

-Even if we assume that the study was well powered to detect the target effect, the reported statistics seem to be wrong, or at least we lack the information to properly reconstruct them. The authors state: "rejection rate in response to the de-value condition was significantly correlated with the degree of dominance within our sample (Pearson's $r = 0.791$, $p < 0.05$)".

Note that no exact p-value is provided in the text. However, it is possible to obtain this information from the N and r values provided. The problem is that, for this coefficient value (0.791) and N (6), the corresponding p-value is 0.061 (assuming a two-tailed test). That is, the results is not significant, contrary to what the paper states. So, if the test was conducted as two-tailed (as is normally by default), then the conclusions are not supported by the data.

-In any case, the authors are providing also a Bayesian analysis to complement the p-value. I appreciate this as I think it can provide useful information. However, I was puzzled as the BF for this analysis is 3.97, which suggests "moderate" evidence for the experimental hypothesis. One of the nice features of the Bayesian approach is that, when information is scarce, the BF produces values pointing to inconclusive results, rather than the false-alarms delivered by p-values. This means that BFs use to be more conservative (with few data) than are p-values. This is what makes the results a bit surprising: with a very small N, we have a nonetheless "high" Bayes Factor, where I would have expected a smaller value.

-I wanted to dig into this issue, but when I downloaded the "raw" dataset provided as supplementary material, I found that the information concerning the social dominance is missing. I think this variable should be part of the shared dataset.

-Then, I reconstructed myself the data from Figure 3. With my reconstructed dataset, $r = 0.792$ (identical to the reported value). The p-value I obtain is 0.061, inconsistent with the results reported. The BF I obtain is also different, $BF = 2.67$, below the cut-off point (a value < 3 is considered as "anecdotal evidence for the hypothesis", equivalent to a nonsignificant p-value). What can explain this discrepancy? I have some possibilities in mind:

First, I thought that the authors were reporting one-tailed tests instead of the standard two-tailed tests, but the numbers still do not coincide.

Second, given the small N, it is possible that they are using a non-parametric analysis that produces a smaller p-value. But then they should clearly explain it in the text (currently, they say “To investigate whether rejection behaviour was correlated with social rank, we used a Pearson’s correlation coefficient”).

Third, the discrepancy with the Bayes factor can be due to a different choice of priors. However, this does not seem to be case as I also conducted a robustness check on my reconstructed data, and no choice of prior in JASP can increase the BF above 2.68.

In sum, after reading the paper and conducting these analyses, I do not know how the results reported in Figure 3 were obtained.

In sum, I think that this is an interesting paper and that the experiment was well conducted. My suggestions to improve the paper are as follows:

- The Introduction needs a better elaboration on how this research is connected to cognitive illusions in humans. I think that the mere reporting of jays using their memory abilities, expectations, and what looks like re-evaluation processes is highly interesting on its own without the reference to illusions.

- The results concerning the dominance variable should be corrected or removed (if they are actually wrong), or completed (if they are right but lack information). With the information that is available, it seems clear that the study was not designed to test this hypothesis with reasonable power.

- The paper needs to incorporate several measures to increase transparency in results reporting. At least: sample size justification, more detail in statistics and how they were computed, and complete dataset available for download.

===PREPARING YOUR MANUSCRIPT===

If you have been asked to revise the written English in your submission as a condition of publication, you must do so, and you are expected to provide evidence that you have received language editing support. The journal would prefer that you use a professional language editing service and provide a certificate of editing, but a signed letter from a colleague who is a native

speaker of English is acceptable. Note the journal has arranged a number of discounts for authors using professional language editing services (<https://royalsociety.org/journals/authors/benefits/language-editing/>).

===PREPARING YOUR REVISION IN SCHOLARONE===

<https://royalsociety.org/journals/authors/author-guidelines/#supplementary-material> to include a suitable title and informative caption. An example of appropriate titling and captioning may be found at https://figshare.com/articles/Table_S2_from_ls_there_a_trade-

off_between_peak_performance_and_performance_breadth_across_temperatures_for_aerobic_sc
ope_in_teleost_fishes_/3843624.

Author's Response to Decision Letter for (RSOS-202358.R0)

See Appendix A.

RSOS-202358.R1 (Revision)

Review form: Reviewer 2

Is the manuscript scientifically sound in its present form?

Yes

Are the interpretations and conclusions justified by the results?

Yes

Is the language acceptable?

Yes

Do you have any ethical concerns with this paper?

No

Have you any concerns about statistical analyses in this paper?

Yes

Recommendation?

Accept with minor revision (please list in comments)

Comments to the Author(s)

I have read the revised version of the manuscript. The authors have included a number of changes in response to the comments from the previous round.

With respect to the use of the term "cognitive illusion", I think it is better explained now.

Concerning the statistical comments, and after correcting a mistake in the dataset, the authors have switched to non-parametric analyses for the correlation with social dominance.

Although I think that this change is for the better, it does not resolve the problem I was pointing out. I was not suggesting that this result in particular was due to failing to meet the assumptions of the statistical model. Rather, I was just observing that the design is underpowered to test this hypothesis in particular, unless we assume that the effect size is huge. I will try to explain myself more clearly.

First, the manuscript includes a practical sample size justification, but no power analysis. Thus, I conducted a sensitivity analysis to help justify the sample size choice. For $N = 6$, the power to detect medium-size correlations ($r = 0.5$) is about 0.20. That is, under the assumption that the

effect is medium-size, only 20% of the time would this experiment yield a significant result. The design (for this particular hypothesis) can only capture reliably (80% of the time) very large effects, $r > 0.85$.

Hence, there is nothing essentially wrong with your inference process. The statistical procedure seems correct as it is the process to reach the conclusions. It just seems that the study was not designed with the aim to test this particular hypothesis with enough power (although it is suitable for the *main* hypotheses). This situation has consequences in terms of reliability and generalization.

To backup my argument, I also computed the 95% confidence interval for the correlation coefficient in this experiment. It goes from 0.21 (small effect) to 0.99 (huge effect). This tells us that the study is probably reporting a true effect, but it gives us little information about its actual effect size.

Consequently, I would just transparently explain in the manuscript that the analysis concerning the social dominance variable should be taken with caution, that the reported estimate is probably larger than the actual value, and that further experiments are needed before settling the question.

Decision letter (RSOS-202358.R1)

Dear Professor Schnell

On behalf of the Editors, we are pleased to inform you that your Manuscript RSOS-202358.R1 "Jays are sensitive to cognitive illusions" has been accepted for publication in Royal Society Open Science subject to minor revision in accordance with the referees' reports. Please find the referees' comments along with any feedback from the Editors below my signature.

Please submit your revised manuscript and required files (see below) no later than 7 days from today's (ie 14-Jul-2021) date. Note: the ScholarOne system will 'lock' if submission of the revision is attempted 7 or more days after the deadline. If you do not think you will be able to meet this deadline please contact the editorial office immediately.

on behalf of Dr Shinya Yamamoto (Associate Editor) and Essi Viding (Subject Editor)
openscience@royalsociety.org

Associate Editor Comments to Author (Dr Shinya Yamamoto):

Associate Editor: 1

Comments to the Author:

Please take the reviewer comment on statistical analysis into consideration, and revise the manuscript accordingly.

Reviewer comments to Author:

Reviewer: 2

Comments to the Author(s)

I have read the revised version of the manuscript. The authors have included a number of changes in response to the comments from the previous round.

With respect to the use of the term “cognitive illusion”, I think it is better explained now.

Concerning the statistical comments, and after correcting a mistake in the dataset, the authors have switched to non-parametric analyses for the correlation with social dominance.

Although I think that this change is for the better, it does not resolve the problem I was pointing out. I was not suggesting that this result in particular was due to failing to meet the assumptions of the statistical model. Rather, I was just observing that the design is underpowered to test this hypothesis in particular, unless we assume that the effect size is huge. I will try to explain myself more clearly.

First, the manuscript includes a practical sample size justification, but no power analysis. Thus, I conducted a sensitivity analysis to help justify the sample size choice. For $N = 6$, the power to detect medium-size correlations ($r = 0.5$) is about 0.20. That is, under the assumption that the effect is medium-size, only 20% of the time would this experiment yield a significant result. The design (for this particular hypothesis) can only capture reliably (80% of the time) very large effects, $r > 0.85$.

Hence, there is nothing essentially wrong with your inference process. The statistical procedure seems correct as it is the process to reach the conclusions. It just seems that the study was not designed with the aim to test this particular hypothesis with enough power (although it is suitable for the *main* hypotheses). This situation has consequences in terms of reliability and generalization.

To backup my argument, I also computed the 95% confidence interval for the correlation coefficient in this experiment. It goes from 0.21 (small effect) to 0.99 (huge effect). This tells us that the study is probably reporting a true effect, but it gives us little information about its actual effect size.

Consequently, I would just transparently explain in the manuscript that the analysis concerning the social dominance variable should be taken with caution, that the reported estimate is probably larger than the actual value, and that further experiments are needed before settling the question.

===PREPARING YOUR MANUSCRIPT===

===PREPARING YOUR REVISION IN SCHOLARONE===

<https://royalsociety.org/journals/authors/author-guidelines/#supplementary-material> to include a suitable title and informative caption. An example of appropriate titling and captioning may be found at https://figshare.com/articles/Table_S2_from_Is_there_a_trade-off_between_peak_performance_and_performance_breadth_across_temperatures_for_aerobic_scops_in_teleost_fishes_/3843624.

Author's Response to Decision Letter for (RSOS-202358.R1)

See Appendix B.

Decision letter (RSOS-202358.R2)

Dear Professor Schnell,

I am pleased to inform you that your manuscript entitled "Jays are sensitive to cognitive illusions" is now accepted for publication in Royal Society Open Science.

on behalf of Dr Shinya Yamamoto (Associate Editor) and Essi Viding (Subject Editor)
openscience@royalsociety.org

Appendix A

Manuscript RSOS-202358

Jays are sensitive to cognitive illusions

Thank you for inviting us to revise this manuscript. We have conducted revisions and believe that our manuscript has greatly improved thanks to the constructive feedback provided by the reviewers – for this we are grateful and thank them for their time and effort. For your reference, we have provided a response below each of the reviewers' comments. We hope that this revised version of the manuscript is well received.

Reviewer comments to Author:

Reviewer: 1

Comments to the Author(s)

A frequent question in comparative cognition is whether two different species of animals are able to successfully address the same environmental challenge. Here the question is inverted: Are humans and Eurasian jay's subject to the same misperceptions (i.e., illusions). Specifically, the manuscript reports a study of Eurasian jays' sensitivity to a slight-of-hand that creates a discrepancy between prior reward experience and current reward for the same response. This parallels humans misperceiving the location of a reward due to slight-of-hand. Observer jays were found to reinspect a receptacle more often when the reward was altered relative to prior experience with that receptacle, and they were slower to consume the food reward when the reward was devalued relative to prior experience. In fact, more dominant jays often completely rejected the devalued reward when their 'expectations' were higher. The report adds to the many illuminating findings concerning avian cognition from Clayton's lab in the past. The descriptions of what was done and what was observed are clear and appropriate.

My one significant concern is that the research was designed to determine whether the jays are subject to the same visual illusion produced by slight-of-hand that humans are. There is a jump from the behavior of the birds to the conclusion that they are subject to the same illusion as humans are. I think that the wording needs to be modified a bit to better differentiate between the observed behavior and the conclusion that the underlying illusion is the same as humans appear subject to. The conclusion at the cognitive level that an illusion was experienced is reasonable, but rests on an assumption concerning the subjects' reasoning to arrive at their behavior. I am not suggesting that this assumption is incorrect, only that there is an assumption here that really goes beyond the data and should be identified as such.

Authors' response: Thanks for this insightful comment. We have now added the following section in the discussion to highlight that the underlying illusion perceived by the jays might be different from the cognitive illusions perceived by humans: "However, this study alone cannot offer conclusive evidence that the magic-switch experiment is exploiting similar cognitive mechanisms in jay observers to those that might be exploited in human observers. Notice that we use an adapted version of cups-and-balls effect and thus the experience of the jays is likely going to differ to that of how humans might experience the traditional cups-and-balls effect. Further

research with comparative analyses are required to pinpoint the similarities and differences between human and non-human observers when perceiving cognitive illusions. Indeed, one recent study suggests that Eurasian jays have different expectations to humans when observing cognitive illusions using sleight of hand techniques [65]. The study found that, similar to humans, jays were deceived by cognitive illusions that utilised fast movements as a deceptive action. By contrast, unlike humans, jays were not deceived by cognitive illusions that capitalised on the observer's expectations with respect to how human hands might manipulate hidden objects [65] (see lines: 410–422).

Of lesser concern:

It is surprising to me that there was so little effort was made to connect the present observations with the extensive literature on successive contrast effect, which ordinarily yields results highly consistent with the present observations. In these earlier reports, devaluation led to reduced effort [a so-called depression effect] akin to the slower consumption observed here.

Authors' response: We agree with Reviewer 1 that our observations show similar patterns to those observed from the negative contrast effect. However, our magic-switch experiment did not use successive trials, which are an important component of negative contrast and positive contrast designs. Thus, it is unlikely that our results could be explained by the contrast phenomenon. For the contrast effect to occur, animals must gain enough information (through reiterate experiences) to create a solid expectation about the proximate future. Specifically, most contrast studies shift from small to large rewards or vice versa after multiple successive trials of one reward type (i.e., 5 - 25 successive trials before switching). By contrast, our experiment proceeded with 12 trials per session, where 3 trials of each of the four conditions (i.e., control 1, control 2, up-value, de-value) were presented to the subjects in pseudo-randomised order. Moreover, the jays did not receive more than two trials of the same condition in successive order, which was conducted to rule-out negative and positive contrast effects. As such, it is unlikely that the observed behaviour in our jays was a result of a 'depression' or 'elation' effects in response to a negative or positive switch because they didn't experience successive trials of one reward type. We have now included these details in the manuscript, both in the introduction 103–108) and the methods (see lines 238–241).

The abstract fails to adequately focus on what was observed and the implications of what was observed in the present study. The first three sentences of the abstract seem superfluous.

Authors' response: Lines 35–40 report on what was observed in the study, and we have added some extra details in the abstract about the implications of our observations (see lines 40–41). However, we have decided to respectfully keep the first three sentences as it provides important context and ecological reasoning as to why the study was conducted.

I wish that “Furthermore, if they are being observed by a competitor, jays will adjust their cache-protection behaviour in response to whether the observer is subordinate or dominant (6).” had been worded such as to tell the reader specifically what had been observed.

Authors' response: We have now elaborated on the patterns observed here and

have added the following sentence: “For instance, jays deprive competitors of visual and acoustic information that might reveal the location of a cache. By contrast, jays refrain from applying these cache protection tactics if they are being watched by their mate with whom they will share their cache at a later time [6].” (see lines 121–124).

Line 3 of Table 2 has an error in it. ‘Stuka was dominant to Stuka’ makes no sense.

Authors’ response: Thank you for bringing this error to our attention. We have now corrected this (see line 674).

Maybe I just failed to see them, but I think that the authors have failed to provide links to their raw data, data analysis, and procedural details.

Authors’ response: All data as well as our statistical analysis outputs are now included in electronic supplementary files.

Reviewer: 2

Comments to the Author(s)

This manuscript reports a study in which jays are presented with a task involving deception: a food item is placed inside a cup, but sometimes it is replaced by a different item (of larger or smaller value) without the bird’s awareness. The finding is that jays produce longer latencies and larger rejection rates for food items that are replacing one of larger value than in the rest of conditions. In general, I think that the research question is interesting, and the study was well conducted. The manuscript describes all the necessary information in a direct and accessible language.

Authors’ response: Thank you for the positive feedback.

-My first comment is conceptual rather than methodological. I am not sure that “cognitive illusion” is the right term to describe what happens in this study. The paper reports a procedure in which expectations are violated by means of a trick. Although I can agree that most (or all) cognitive illusions involve expectation violation to some extent, I do not think that all paradigms that involve violation of expectations can be considered “a cognitive illusion”. Certainly, I am not sure that I would use this word if I were observing a similar procedure to the one in this experiment, but in humans. The Introduction section would improve if additional detail is given as to how this study connects with the typical scenarios in which we talk about “cognitive illusions” (in humans). I do acknowledge, however, a subtle but reasonably well explained connection to the concept of “cognitive bias”, as jays seem to fall prey of the well-known “loss aversion” bias.

Authors’ response: Thanks for this insightful comment. We agree that not all violation of expectation paradigms can be considered a cognitive illusion. However, we respectfully have decided to keep the term cognitive illusion in the paper because the ‘cups-and-balls’ routine is often described as a magic illusion/cognitive illusion that manipulates human cognition and perception. We now include this information in the introduction to help justify why we use the term cognitive illusion throughout this study: “The cups-and-balls routine is typically described as a cognitive illusion [41] because it is used to manipulate human perception and cognition by creating an illusion in which balls ‘magically’ appear and disappear under the cover of opaque

cups [42,43]. In humans, the cognitive mechanisms underlying this well-known illusion has been extensively studied by psychologists who have aimed at disentangling the epicentral elements of the experience [41,43,44]. Here, for the first time, we adapt this illusion to investigate the response in non-human animals. The application of this illusion allowed us to not only test whether jays are sensitive to the content of hidden items but also whether their responses are augmented or diminished depending on how the content is changed” (see lines 128–136).

-The sample size is small (initial N=9, final N=6). This can be fine in many situations, specially in procedures like these that involve nonhuman animals and multiple measures. However, a small N definitely limits the conclusions one can draw from data. In line with current practices to increase transparency, I would like to read in the paper a justification on how the sample size was decided. Ideally, we would have (1) a target effect size from previous studies, or (2) a sensitivity analysis that ensures we can observe meaningful effects with reasonable power (i.e., you can fix N and power, and report the smallest effect size you can detect).

As Reviewer 2 notes, small sample sizes are not uncommon in comparative psychology – a recent review paper reported that out of 116 experiments on physical cognition, the median sample size was 7 (Farrar et al. 2020)¹. As is the case in many comparative psychology studies (Lakens, 2021)², we were limited by resource constraints. In our study, all participation was voluntary, and we tested the maximum number of jays in our lab that would voluntarily participate in the experiment. To ensure we could maintain statistical power we tried to increase trial number per condition per subject (Rouder & Haaf, 2018)³. Out of the 16 birds from our colony, we often have $n = 10$ that will voluntarily fly into the testing compartment and interact with the apparatus and the experimenter. However, participation can diminish depending on the type of experiment that is being conducted. In our case, three of our original sample refused to participate after being exposed to the de-value condition over several trials. Given that this experiment was novel and was not a replication of a previous study, we had no notion of an effect size and distribution of the effect and thus we were unable to estimate power a priori. The best we could do was estimate power for some particular distribution we thought might be close. We predicted that our data would not meet the assumptions of normality because data obtained from animal cognition studies often do not match the assumptions of normality. Thus, we expected we would use non-parametric permutation tests, which can provide exact control of false positives and can be used to detect differences in sample > 5 .

¹ Farrar, Altschul, Fischer, van der Mescht, Placi, Troisi, Vernouillet, Clayton & Ostojić. (2020). Trialling meta-research in comparative cognition: Claims and statistical inference in animal physical cognition. *Anim. Behav. Con.* 7; 419–444.

² Lakens. (2021). Sample size justification. PsyArXiv (preprint server)

³ Rouder & Haaf. (2018). Power, dominance, and constraint: A note on the appeal of different design traditions. *Assoc. Psychol. Sci.* 1; 19–26.

-Also to improve transparency, I would recommend to include all test statistic values and the rest of information necessary to interpret them (e.g., degrees of freedom), as in many analyses we can only see the p-values and effect sizes.

Authors' response: We apologise for not reporting all test statistics in the results section of the manuscript. These have now been reported as F statistics, degrees of freedom and p-values.

-On a similar note, it is recommended to report exact p-values, instead of the old phrase “ $p < 0.05$ ”.

Authors' responses: For readability and clarity we have continued to use the method of reporting $p < 0.05$; 0.01 ; 0.001 within the main results section of the text but we now also include an electronic supplementary file with all statistical analysis outputs from both RStudio and JASP, which reports exact p-values.

-But the most worrying issues concern the dominance analysis in Figure 3. First, when looking at the figure, it makes the reader wonder whether the analysis is correct. Specifically, with a N of 6 (as this analysis does not contain multiple measures), is it right to compute a Pearson's r? Is this estimation reliable? I think that, unless more information is made available, the answer is that the study lacks power for testing this hypothesis.

Authors' response: Thank you for pointing this out. After consideration, we agree with Reviewer 2 that the Pearson's r is not the most appropriate analysis given our small sample size, so we have decided instead to use a two-tailed Kendall's tau-b correlation coefficient instead. Long and Cliff (1997) found that Kendall's tau performed well with small samples < 25 . Nevertheless, given that we have a sample size of 6, we continue to use a two-tailed Bayesian correlation analysis to determine the strength of the evidence in support of our experimental hypothesis.*

-Even if we assume that the study was well powered to detect the target effect, the reported statistics seem to be wrong, or at least we lack the information to properly reconstruct them. The authors state: “rejection rate in response to the de-value condition was significantly correlated with the degree of dominance within our sample (Pearson's $r = 0.791$, $p < 0.05$)”.

Note that no exact p-value is provided in the text. However, it is possible to obtain this information from the N and r values provided. The problem is that, for this coefficient value (0.791) and N (6), the corresponding p-value is 0.061 (assuming a two-tailed test). That is, the results is not significant, contrary to what the paper states. So, if the test was conducted as two-tailed (as is normally by default), then the conclusions are not supported by the data.

Authors' response: Apologies for the confusion, we have found a computational error in our formatted CSV file that was used for the analysis and have now corrected this using our Master data file. All data as well as the analysis outputs from both RStudio and JASP are now also included as electronic supplementary files so that the readers have access to all necessary information and our statistical analysis.

* Long, J.D., & Cliff, N. (1997). Confidence intervals for Kendall's tau. *British Journal of Mathematical and Statistical Psychology*, 50, 31–41.

-In any case, the authors are providing also a Bayesian analysis to complement the p-value. I appreciate this as I think it can provide useful information. However, I was puzzled as the BF for this analysis is 3.97, which suggests “moderate” evidence for the experimental hypothesis. One of the nice features of the Bayesian approach is that, when information is scarce, the BF produces values pointing to inconclusive results, rather than the false-alarms delivered by p-values. This means that BFs use to be more conservative (with few data) than are p-values. This is what makes the results a bit surprising: with a very small N, we have a nonetheless “high” Bayes Factor, where I would have expected a smaller value.

Authors' response: As mentioned above, we have found an error in our CSV file that was used to calculate the rejection rate proportions. We have now corrected this and include all data and our analysis outputs as electronic supplementary files. With this correction in place, our data shows that there is moderate evidence in support of our experimental hypothesis.

Appendix B

Manuscript RSOS-202358

Jays are sensitive to cognitive illusions

Thank you for accepting our manuscript. We have responded to the reviewer's comment below. We look forward to being published in the Royal Society of Open Science.

Reviewer comments to Author:

Reviewer: 2

Comments to the Author(s)

I have read the revised version of the manuscript. The authors have included a number of changes in response to the comments from the previous round. With respect to the use of the term “cognitive illusion”, I think it is better explained now. Concerning the statistical comments, and after correcting a mistake in the dataset, the authors have switched to non-parametric analyses for the correlation with social dominance. Although I think that this change is for the better, it does not resolve the problem I was pointing out. I was not suggesting that this result in particular was due to failing to meet the assumptions of the statistical model. Rather, I was just observing that the design is underpowered to test this hypothesis in particular, unless we assume that the effect size is huge. I will try to explain myself more clearly. First, the manuscript includes a practical sample size justification, but no power analysis. Thus, I conducted a sensitivity analysis to help justify the sample size choice. For $N = 6$, the power to detect medium-size correlations ($r = 0.5$) is about 0.20. That is, under the assumption that the effect is medium-size, only 20% of the time would this experiment yield a significant result. The design (for this particular hypothesis) can only capture reliably (80% of the time) very large effects, $r > 0.85$. Hence, there is nothing essentially wrong with your inference process. The statistical procedure seems correct as it is the process to reach the conclusions. It just seems that the study was not designed with the aim to test this particular hypothesis with enough power (although it is suitable for the *main* hypotheses). This situation has consequences in terms of reliability and generalization.

To backup my argument, I also computed the 95% confidence interval for the correlation coefficient in this experiment. It goes from 0.21 (small effect) to 0.99 (huge effect). This tells us that the study is probably reporting a true effect, but it gives us little information about its actual effect size.

Consequently, I would just transparently explain in the manuscript that the analysis concerning the social dominance variable should be taken with caution, that the reported estimate is probably larger than the actual value, and that further experiments are needed before settling the question.

Authors' response: We thank Reviewer 2 for the clarification. We have taken Reviewer 2's recommendation on board and have added the following sentence in the discussion section – ‘Nevertheless, consider the reported correlation statistics with caution since our limited sample size could lead to an underpowered correlation analysis. Replications of the current study with a larger sample size might highlight

finer-grained analyses and could improve the evidential value of our findings' (see lines 424–427).